# Mapping Asia Plants: Historical Outline and Review of Sources on Floristic Diversity in South Asia

**DOI:** 10.3390/plants12081617

**Published:** 2023-04-11

**Authors:** Cui Xiao, Zhixiang Zhang, Keping Ma, Qinwen Lin

**Affiliations:** 1School of Ecology and Nature Conservation, Beijing Forestry University, Beijing 100083, China; 2State Key Laboratory of Vegetation and Environmental Change, Institute of Botany, Chinese Academy of Sciences, Beijing 100093, China; kpma@ibcas.ac.cn; 3National Botanical Garden, Institute of Botany, Chinese Academy of Sciences, Beijing 100093, China

**Keywords:** South Asia, flora, checklist, biodiversity, Mapping Asia Plants (MAP)

## Abstract

South Asia, which is composed of eight countries, including Afghanistan, Bangladesh, Bhutan, India, Maldives, Nepal, Pakistan, and Sri Lanka, is an important global biodiversity hotspot. As a part of the Mapping Asia Plants (MAP) project, we reviewed the history of botanical investigations, floristic works, and publications in this region, as well as the key floras, checklists, and online databases in South Asia. The botanical survey of this region, which began during the 17th century, has two distinct phases: surveys conducted during the British India period and those conducted in the post-British period. The seven volumes of *The Flora of British India* are the most important contributions to flora research in South Asia because of their wider geographical coverage, which was performed by British botanists. Following on from this, different countries have launched independent floristic surveys. At the country level, Afghanistan, Bangladesh, Bhutan, India, Nepal, Pakistan, and Sri Lanka have completed, or partially completed, their flora surveys at the country level, while Maldives has not yet published its national flora survey. According to currently available information, the approximated numbers of plant taxa for each country in South Asia are as follows: Afghanistan, 5261 (vascular plants); Bangladesh, 3470 (vascular plants); Bhutan, 5985 (flowering plants); India, 21,558 (flowering plants); Maldives, 270 (common plants); Nepal, 6500 (flowering plants); Pakistan, 6000+ (vascular plants); and Sri Lanka, 4143 (flowering plants). Additionally, there are 151 books devoted to the key floras and checklists in South Asia. A total of 1.1 million digital records of specimens from this region can be found on the website of the Global Biodiversity Information Facility (GBIF). However, there are still major gaps and limitations—such as out-of-date publications, national floras that are mainly detailed only in local languages, massive non-digitized specimens, and the lack of a comprehensive online database or platform—which should be addressed in terms of their global applications.

## 1. Introduction

Recent advancements in biodiversity informatics have led to significant improvements in global and regional-level biodiversity databases [1]. The abundance of species distribution data has provided robust support for biodiversity research and greatly facilitated the study of large-scale biodiversity patterns, conservation planning, responses to global changes, and prediction of introduced species’ invasive potential [2]. However, as a whole, Asia is a data-poor region, and there is also no regional-scale biodiversity database. Although China, India, Japan, and Korea have established foundations in biodiversity database construction [3], many other Asian countries lack complete biodiversity databases. To address this issue, the Asian Biodiversity Conservation and Information Network (ABCDNet) adopted the Chinese-proposed Mapping Asia Plants (MAP) program for digitizing plant diversity in Asia at its annual working meeting in November 2015. The proposed Mapping Asia Plants (MAP) program aims to establish an online platform for the collection, organization, and dissemination of big data on Asian botanical information. This platform aims to provide a comprehensive and interdisciplinary data-mining environment for the conservation and research of Asian plant diversity [4]. The initial phase of the program focuses on the creation of a database that compiles the list and distribution of vascular plant species in Asia. Under MAP, Asia is divided into six sub-regions: Southeast Asia, South Asia, West Asia, Central Asia, North Asia (i.e., the Russian portion of Asia), and Northeast Asia [5,6]. Each region is overseen by a designated member whose primary responsibility is to gather information on the lists of plant species and their distribution within their respective area.

South Asia, also known as the Indian subcontinent, is a region comprising eight countries: Afghanistan, Bangladesh, Bhutan, India, Maldives, Nepal, Pakistan, and Sri Lanka (Figure 1). It covers an area of 4.4 million square kilometers, or 10% of the continent, and is home to over 24% of the world’s population, making it the most densely populated region on Earth [7,8]. It is a diverse region with a wide range of natural landscapes and ecosystems [9]; it is characterized by rugged mountain ranges, including the Hindu Kush, the Karakoram, and the Himalayas in the north. The region is also known for its vast plain areas, including the Indo-Gangetic Plain, which is formed by the combined alluvial plains of the Indus, Ganges, and Brahmaputra rivers [9]. South Asia also boasts a variety of vegetation types, including tropical rainforests, mangrove swamps, deciduous forests, alpine scrubs, and meadows. These vegetation types are home to a wide variety of plant and animal species. The region is also home to a number of large rivers, including the Ganges, Brahmaputra, and Indus, as well as many other smaller rivers and streams.

South Asia also has important global biodiversity hotspots, including the mountain ranges of the Himalayas and Hindu Kush, the Western Ghats of India, and the islands of Sri Lanka [10]. It has a biological richness and diversity that is unequaled. The botanical survey of this area began during the 17th century, and *The Flora of British India* is perhaps the most important contribution to the flora of the region [11,12]. For the period after the dissolution of British India, the botanical activities of eight countries in the region gradually developed in their own ways and became different. To date, a comprehensive and integrated catalog of plant diversity data remains elusive. The current study aims to provide an overview of the historical development of botanical research in South Asia, highlighting significant contributions from individual countries within the region. Additionally, an analysis of the current state of plant diversity data in the region is presented.

## 2. A Brief History of Floristic Research in South Asia

The history of botanical surveys in South Asia can be divided into two periods: one corresponding to the British India period, and the other corresponding to post-British India, i.e., after 1947. During the period of British India, King reviewed most of the history of botany that is detailed [13].

### 2.1. Early Period Exploring of South Asia

The initial contributions to the botanical knowledge of the region that is now known as British India were made by Dutch botanists. The first was the ‘*Hortus Malabaricus*’, which was published between 1678 and 1693 and illustrated with 794 plates from the Malabar region of Kerala [13]. This work was compiled by Hendrik Adriaan van Reede tot Drakenstein and features captions in five different languages [12]. Another significant contribution was the ‘*Herbarium Amboinense*’, published between 1741 and 1755, which comprised 7 folio volumes and 969 plates [14]. This work was based on the collections of George Everhard Rumph and was published by Professor John Burman. The ‘*Thesaurus Zeylanicus*’, which was founded on the collections of Paul Heremann from his seven-year exploration of the *Flora of Ceylon* (1670–1677), was published in 1737 by John Burman [15].

Following on from this, the classic history of botanic science in the main part of South Asia can be divided into two periods, the first extending from J.G. Koenig’s arrival in India in 1768 to Sir Joseph Hooker’s arrival in 1848, and the second period being from the latter date (i.e., 1848) to the dissolution of British India (i.e., 1947) [13]. Within this period, the most important work that must be considered is Joseph Dalton Hooker’s *The Flora of British India*, which was published in seven volumes between 1872 and 1897 [11].

As mentioned prior, the first period began in 1768 as J.G. Koenig, a pupil of Linnaeus, arrived in India. He began the modern study of botany with the fervor of an enthusiast that he succeeded in imparting to various correspondents, who were then settled near him in Southern India [16]. They formed a society named ‘*The United Brothers*’ to promote botanical study and began sending specimens to Europe. Many plants of Indian origin came, thus, to be described by Retz, Roth, Schrader, Willdenow, Vahl, and Smith [11]. After those, Dr. William Roxburgh was the pioneering botanist who sought to compile a systematic account of plants in India. His book ‘*The Plants of the Coast of Coromandel*’ (1795–1818), the manuscripts of ‘*Flora Indica*’ [17,18], the ‘*Hortus Bengalensis*’, and his remarkable colored drawings (mostly of natural size)—which featured 2533 species of indigenous Indian plants—all serve as the foundation for subsequent works on Indian botany [19]. Robert Wight made significant contributions through his works, including the ‘*Illustrations of Indian Botany*’ (1840), the ‘*Icones Plantarum Indiae Orientalis*’ (1838–1853), and the ‘*Prodromus Florae Peninsulae Indiae Orientalis*’ (1834) in which the natural system of classification was employed and several new taxa were described [20]. William Griffith (1808–1866) explored Tenasserim, the Assam valley, Mishmi, Kasia, the Naga ranges, the Hookung valley, the Nerbudda valley, the Bhutan, Sikkim, Kabul, Khorassan, Simla, Irrawaddy, Rangoon, and Malacca. He collected more than 9000 species and documented numerous descriptions of plants from living specimens, many of them being from the South Asian region [13].

Notable contributions during the 19th century include the ‘*Illustrations of the Botany and other branches of the Himalaya Mountains: and of the flora of Cashmere*’ [21], ‘*The Bombay Flora*’ [22], the ‘*Hortus suburbanus Calcuttensis*’ [23], ‘*The Flora Sylvatica for Southern India*’ [24], the ‘*Icones Plantarum Indiae Orientalis* [20], and the ‘*Flora Indica*’ [17,18].

Given that Kashmir is a disputed region between India and Pakistan, the information about botanical surveys regarding this region may be more appropriate to mention here. The history of plant collecting in Kashmir began with a small collection by William Moorcroft in 1820. The first person to enter Kashmir proper was Victor Jacquemont. The first man who was interested in Kashmir plants was J.F. Royle, who was in charge of a botanical garden at Saharanpur in the United Provinces and had many Kashmir plants in cultivation. Keshavanand collected plants in Kashmir in 1906–1909 and was one of the first local collectors. T.S. Sabnis of the organization *Survey of India*, who collected plants in Sind and the Punjab, was one of the first Indians to work in this area for the survey [25].

The inception of the second period of the history of South Asian floristic research can be traced back to 1848 with the arrival of Joseph Hooker in India. J.D. Hooker’s publication of ‘the *Flora of British India*’ ran during 1872–1897, and it is a seven-volume work that encompasses the phanerogams of former British India. Furthermore, it is a significant milestone in the study of the flora of the region [11]. This publication was followed by the ‘*Flora of the Presidency of Madras*’ [26]. Then, there was the first botanical publication on the plants of the Western Ghats, ‘*Coloquis dos Simples*’ [27], which is a checklist of medicinal plants in India and which marks the beginning of botanical studies in this region. A century later, a more comprehensive work on the flora of the Malabar coast, the ‘*Hortus Indicus Malabaricus*’ [28], was published. The British botanists made significant contributions to the floristic studies in India, among whom Robert Wight’s contributions stand out as the most remarkable [29]. 

In addition to these works, the ‘*Flowering Plants of Travancore*’ [30], the ‘*Flora of Anamalai Hills, Coimbatore District, Madras Presidency*’ [31], and the ‘*Flora of South Indian Hill Stations*’ are some of the other notable works on the flora of the region [32]. The ‘*Forest Trees of Travancore*’ [33], the first comprehensive work on the tree flora of Travancore, which deals with 582 indigenous trees, is also worth mentioning.

### 2.2. Floristic Exploring at Country Level

For the period after the dissolution of British India, the countries in South Asia gained independence successively, and their botanical activities gradually developed in their own ways and became different.

### 2.3. Afghanistan

Afghanistan is rich in plant species because of the greater diversity of its habitats, ranging from hot deserts and humid subtropical regions to high alpine regions. The exploration of Afghanistan flora started in the early 19th century when three Anglo-Afghan wars brought not only troops, but also scientists such as Moorcroft (1830). Furthermore, Griffith (1840) published the ‘*Itinerary Notes of Plants Collected in the Khasyah and Bootan Mountains* 1837–1838’ in 1848, and Aitchison (1880) published ‘*On the Flora of Kuram Valley*’ [34].

The relationship between Afghanistan and Germany dates from 1923 and the German-sponsored Nedjat or Amani School in 1924, which was when many German teachers made botanical collections there, such as Volk who published ‘*Klima und Pelanzenverbreitung in Afghanistan*’ [35]. After the Second World War, the ‘*Symbolae Afghanicae*’ was edited by K.H. Rechinger with various contributors to its six parts, which were published between 1954 and 1965 [36]. In 1963, the first slender Araceae ‘Flora Iranica’ fascicle by K.H. Rechinger (Vienna) was published, which consists of some 172 fascicles [37]. In February 1966, the Scientific Afghanistan Research Group (AGA) was founded by Prof. Dr. Willy Kraus and Prof. Dr. Carl Rathjens during a co-ordination meeting at the South Asia Institute. Between 1974 and 1982, the ‘Afghanistan Journal’ was published [13]. Between 1960 and 1980, a great number of relevant scientific publications were published by expedition and mountaineering teams who were active throughout Afghanistan [38]. In 2002, Prof. Clas Naumann founded the German-Afghan University Association (DAUG) in order to help Afghan universities in a more flexible way [39]. In 2012, the ‘*Checklist of the Flowering Plants of Afghanistan*’, an important checklist of Afghan plants, was published by D. Podlech, who is a full professor of plant systematics at the Ludwig Maximilian University of Munich, Germany [40]. This also became the reference basis for subsequent related works.

In 1960, a Japanese botanist published the ‘*Flora of Afghanistan*’, which is, in actuality, a synoptically and unsystematically constructed checklist and contains the results of the Kyoto University Scientific Expedition to Karakorum and the Hindu Kush in 1955 [41].

Russian botanists have also collected and published on various facets of Afghanistan’s plants. Their publications can be found in the ‘*Botanical Literature of Afghanistan*’ and its supplement [42,43]. 

The ‘*Vascular Plants of Afghanistan: an Augmented Checklist*’ was published in 2013 [39] and is currently the most complete and comprehensive list of Afghan plants; this publication is also partly a natural follow-on from the earlier ‘*Field Guide Afghanistan—Flora and Vegetation*’, which was published in 2010 by two of the same authors, S.-W. Breckle and M.D. Rafiqpoor [44].

Therefore, the floristic research in Afghanistan is based almost entirely on the collections made by botanists from many different countries, mostly European, and a few who were from Afghanistan.

### 2.4. Bangladesh

Today’s Bangladesh is only a small part of former British India. In addition, it was also once the Bengal provinces of Pakistan; as such, its early flora-related research works were usually included in the works of the Indian subcontinent at that time. After the independence of Bangladesh in 1971, flora-related works were conducted mainly by Professor Salar Khan, from the University of Dhaka, and his students. Professor Khan and his students presented several excellent works, such as the *Flora of Bangladesh* (1972–2015) [45], the *Timber Plants of Bangladesh (Wild and Cultivated)* (1986), the *Aquatic Angiosperms of Bangladesh* (1987), the *Red data book of Vascular plants of Bangladesh* (Vol. 1 2001, Vol. 2 2013), the *Bulletin of the Bangladesh National Herbarium* (since 1988), and the *Encyclopedia of Flora and Fauna of Bangladesh* (Vol. 5–12, 2007–2009) [46,47,48,49,50,51,52,53]. 

### 2.5. Bhutan

Bhutan was included in the *Flora of British India* (1875–1897) as a result of some 1200 plants that were collected in 1838 by William Griffith, the first botanist to visit Bhutan [11]. Bhutan is a complex mixture of culture, religion, and politics. Bhutan maintained its independence and allowed very few visitors to enter the country until the 1970s. It contained, in total, only 17 botanical explorations before 1975. From 1914, a small number of privileged British botanists and horticulturists were able to travel extensively in Bhutan. Frank Ludlow and George Sherriff made seven visits to Bhutan between 1933 and 1950, and collected over 6000 specimens [54]. Grierson and Long visited Bhutan in 1975, 1979, and 1982, and collected over 2500 specimens. Simon Bowes Lyon visited the country on many occasions, arising from his friendship with the Bhutanese royal family. He collected many specimens in 1966, 1967, 1969, 1971, and 1994, which are now located mainly at the British Museum. John R.I. Wood made the most important individual contribution to our knowledge of the flora after Griffith. He was an outstandingly gifted and observant botanist with wide experience. He worked in the Department of Education from 1987 to 1992 and collected some 1800 specimens. Bhutanese started taking an interest in the flora of their country from the 1970s; the first of whom were R. Nawang and Sonam Tshering. 

Although there are accounts of travel journals and papers published by botanists visiting Bhutan, significant floristic works did not exist until the *Flora of Bhutan* [54,55,56,57,58,59,60,61,62]. The number of specimens collected from Bhutan is estimated to be in excess of 26,600. (https://www.gbif.org/occurrence/search?basis_of_record=PRESERVED_SPECIMEN&country=BT&occurrence_status=present (accessed on 13 February 2023)).

### 2.6. India

*The Flora of British India,* published in seven volumes by J.D. Hooker and his associates (1872–1897), is the only standard reference work on which botanists in India rely upon to study the flora of the country [11]. The area covered by Hooker’s work has undergone political transformation following the end of the colonial period. In a long period after the dissolution of British India, the organization, the *Botanical Survey of India* (BSI), has been continuing the work of surveying, exploring, inventorying, documenting, and conserving the wild plant diversity of India. Up to now, there have been many works published by the BSI, including annual checklists of the plants discovered (new species and new records) in India for many years. The project to publish a new version of the *Flora of India* has also been conducted by the BSI, and 11 volumes have already been published since 1997 [63]. Unfortunately, an integrated national flora record is still unavailable today, no matter how eagerly it is needed for many reasons. 

Up until now, there are ca. 49 sets of flora in the states of India that have been published, such as the *Flora of Himachal Pradesh-Analysis* [64,65,66,67,68,69,70,71,72,73,74,75,76,77,78,79,80,81,82,83,84,85,86,87,88,89,90,91,92,93,94,95,96,97,98,99,100,101,102,103,104,105,106,107,108,109,110,111,112]. Some state-level flora recordings are just now getting started, most of which have not been completed, and almost all of the works are based on George Bentham’s and Joseph D. Hooker’s system or the *Flora of British India*. However, one thing is very positive, that is, almost all the local flora is written in English [113].

### 2.7. Maldives 

There are limited botanical works regarding Maldives because there are limited numbers of plant species due to its harsh environmental conditions for the growth of plants. With its tropical climate and covering an area of 90,000 km^2^, Maldives is actually made up by 1190 coral islands, most of which are very small (and which vary in size between 0.5 and 5 km^2^), are flat, and are without hills or rivers. Nearly 80% of the land area is less than 1 m above the mean high tide level, and the average elevation above the mean high tide level is only ca. 1.8 m. The soil of Maldives is saliniferous, nutrient poor, and unsuitable for plant growth. Therefore, there are only a few plant species recorded and only a few botanical explorations and literature in Maldives. Selvam (2007) published a book, the *Trees and Shrubs of the Maldives*, which contains information on 100 woody plant species in Maldives [114]. Furthermore, Sujanapal and Sankaran (2016) contributed a new book, the *Common Plants of Maldives*, which includes information on 270 species of both native and introduced vascular plants in Maldives [115]. There is also a database on biodiversity information (the *Maldives Materia Medica Database*, http://www.biodiversity.com.mv/eng/ (accessed on 13 February 2023)) that is available online, which includes more than one thousand records of plants in Maldives, although many of these plants were introduced and cultivated.

### 2.8. Nepal

The botanical exploration of Nepal began in 1802 when the East India Company sent Francis Buchanan, a Scottish medical man in the company’s employment, to Kathmandu in order to explore Nepalese plants. He stayed in Nepal from March 1802 to March 1803 under trying conditions and assembled a valuable collection of dried plants [116]. Wallich went to Kathmandu in 1820 and stayed there for one year. He described new Nepal plants in his own collections, which were more extensive than Buchanan’s and, for a long period, were the only Nepal specimens available to botanists [117]. There was a long gap in botanical collection within Nepal following the visits of Buchanan-Hamilton and Wallich because of political reasons. It was not until 1848 that J.D. Hooker made a winter journey into eastern Nepal from Sikkim, occupying the region for three months. In addition, the Schlagintweit brothers in 1857 and J. Scully in 1876 also visited Kathmandu. Additionally, it was not until 1907 that another botanist, I.H. Burkill, visited Nepal and once more followed Wallich’s route to Kathmandu [118]. 

Nevertheless, no major botanical collection came out of Nepal from 1908 until 1927–1937, which was when Lall Dhwoj and then N.K. Sharma completed their collections of herbarium specimens and seeds for presentation to King George V. Following on from this, there was again a long gap in collection and Nepal remained a relatively closed country to European plant collectors until 1949 [118]. After 1949, there were many mountaineering parties, who attended with naturalists and went to Nepal for botanical exploration, e.g., Polunin and Lowndes in 1949 and 1950. The stimulus for the exploration of plants in Nepal came especially through the influence and activity of F. Ludlow and G. Taylor [119].

Major botanical expeditions then followed. The most important individual contribution, covering the whole country, was that of J.A. Stainton, whose journeys in 1954 and 1956 every year, from 1962 to 1971, and in 1974 and 1975 covered the whole country. Stainton’s excellently prepared material included many new species and even new genera. From these activities, the Natural History Museum (once a part of the British Museum) in London acquired around 50,000 Nepalese specimens [117].

Between 1960 and 1972, the University of Tokyo organized some botanical expeditions from eastern Nepal to Sikkim and to Bhutan under the leadership of H. Hara, who completed rich collections that are now preserved in Tokyo [118,119,120]. The Nepal Government established a herbarium in 1960 in its Department of Medicinal Plants at Kathmandu. P.N. Suwal, who served as the Director, and S.B. Malla, who was the Acting Director, have done much to further the botanical work there and to support collecting expeditions [117].

With 218 years of explorations to date, there have been a number of additions of species, as well as new discoveries. Based on the publication of new plants in various research articles, journals, books, and on the herbarium specimens of type collection, Rajbhandari et al. (2019) presented 1600 new flowering plants, which were discovered from Nepal, in the publication the ‘*Flowering plants discovered from Nepal*’ [121]. Shrestha and Press, Rajbhandari, and Yano also provided information on the type specimens of Nepalese flowering plants [122,123].

### 2.9. Pakistan

There was little known of the plants of Pakistan until after the Punjab was taken over by the British after their defeat of the Sikhs in 1848–1849. In 1835, Honigberger was in search of plants of medicinal value, crossed into the Punjab, and was able to complete many collections. Dr. William Griffith collected plants in the Punjab, Sind, and Baluchistan on his 1839–1840 campaign. Later, explorers and mountain climbers from all over the world came to Pakistan. The scientific officers of the Government of India, such as Hugh Falconer, J.E. Stocks, William Griffith, J.F. Duthie, Thomas Thomson, and C.B. Clarke, also completed a great deal of collections. Much of the great collection of J.R. Drummond from the Punjab was gathered by local collectors. When Indian foresters began to be trained at Dehra Dun Forest School (which is now the Forest Research Institute, Dehradun), some of them went on to become collectors [124]. 

With the division of India in 1947 and the departure of most of the foreign scientists, Indian and Pakistani botanists started taking over the leading role in terms of systematic work. For many years before 1947, there was an active Department of Botany at the Punjab University, Lahore, under the able leadership of Prof. Shiv Ram Kashyap. He collected in Ladak and, together with Prof. A.C. Joshi, published a Lahore District Flora in 1936 [125]. Several keen Pakistani botanists have come to the fore since 1947, such as Dr. S.I. Ali of Karachi, who has been already mentioned. In 1960, under the Columbo Plan, Mr. B.L. Burtt of Edinburgh was sent to Pakistan for a year to study the likelihood of starting a National Herbarium for Pakistan [126]. Later, the largest collection of the Stewart Herbarium at the Gordon College in Rawalpindi was presented as a gift to the nation, and it formed the nucleus of the National Herbarium of Pakistan (ISL) at the Quaid-i-Azam University, containing ca. 180,000 specimens of plants (https://www.gbif.org/dataset/6f40c8ec-2b95-47d5-9064-eda6a8ddf3fb#description (accessed on 13 February 2023)). There are smaller collections in Peshawar, Lahore, Lyallpur, and Karachi.

### 2.10. Sri Lanka 

Sri Lanka is an island with a relatively independent biogeography with many endemic species. This fact has long been of great interest for botanists. As an exception, Henry Trimen published the *Handbook to the Flora of Ceylon* during 1893 to 1900, which was a good integrated flora record for its time, but is now very old, unobtainable, and outdated [127]. Following on from this, after some decades, the series of *A Revised Handbook to the Flora of Ceylon* (Volume 1–15) were gradually published under the PL-480 support by USA in 1980–2006 [128]. In addition, under the support of the National Science Foundation of Sri Lanka, Lilani Kumudini Senaratna (2001) published *A Check List of the Flowering Plants of Sri Lanka*, which was based primarily on *A Revised Handbook to the Flora of Ceylon* and certain other works [128,129].

## 3. Regional and Country-level Floras and Checklists on the Plants of South Asia

### 3.1. The Regional Level—South Asia

#### *Flora of British India*, Vol. 1–7 (1872–1897)

The *Flora of British India* by J.D. Hooker and his collaborators was published in English, with keys to genera but not species [11]. Synonymy, references, citations, etc., are also presented, but the distribution is often quite vague, where only large geographical names are mentioned. The publication presents approximately 14,300 species (171 families, 2325 genera) of flowering plants. In British India, this included flora and comprised not only six countries in South Asia (Afghanistan, Bangladesh, India, Nepal, Pakistan, and Sri Lanka), but also other regions, including Myanmar, Xizang of China, and the Malayan Peninsula. The *Flora of British India*, along with other local floras, served taxonomic works well until the end of the 20th century [130].

### 3.2. Afghanistan

#### 3.2.1. *The Flora of Afghanistan* (1955) 

The results of the Kyoto University Scientific Expedition to Karakoram and the Hindu Kush in 1955 included seven separate volumes dealing with the botanical, zoological, anthropological, and geological aspects of the expedition. The plant volumes were published with the assistance of the Fauna and Flora Research Society, Kyoto University, in 1960 [131]. The *Flora of Afghanistan* lists 3100 species, but with no family nor general information [41]. However, the accepted names, synonyms, distribution, etc. are encompassed within it.

#### 3.2.2. *Checklist of the Flowering Plants of Afghanistan* (2012)

This checklist of all the flowering plants that were hitherto unknown from Afghanistan includes 4482 taxa [40]. The checklist is based on the *Flora Iranica* [37], on several new taxonomic works of different groups, and on the rich material housed in the herbaria M and MSB of München in Germany, and herbaria W of Wien in Austria.

#### 3.2.3. *Vascular Plants of Afghanistan: An Augmented Checklist* (2013)

This book comprises 149 families, 1086 genera, and 4826 species. The 5035 taxa provide the most accurate and up-to-date account of the plant diversity of Afghanistan [132]. It complements the colorful ‘*Field Guide Afghanistan: Flora and Vegetation*’, which was published in 2010 during the International Year of Biodiversity, and was funded by the German Academic Exchange Service (DAAD) [44]. This book represents an excellent way by which taxonomists can understand the flora of Afghanistan.

### 3.3. Bangladesh 

#### 3.3.1. *Encyclopedia of Flora and Fauna of Bangladesh*, Vol. 5–12 (2007–2009)

In total, there are 28 volumes of the *Encyclopedia of Flora and Fauna of Bangladesh*. These were published by the Asiatic Society of Bangladesh during 2007–2009 [46,47,48,49,50,51,52,53]. In addition, the works for vascular plants are found in volume 5 (pteridophytes and gymnosperms) and volumes 6–12 (angiosperms). These volumes represent a record of full descriptions and images of all the plant species that were discovered in Bangladesh before the publishing of the Encyclopedia. There are 196 species of pteridophytes, 7 species of gymnosperms, and 3267 species of angiosperms in the Encyclopedia. It could, thus, be regarded as an accurate record of the distinct and excellent modern flora among the countries of South Asia.

#### 3.3.2. *Flora of Bangladesh* (Vols. 1+, 1972+)

The *Flora of Bangladesh* is an irregularly published series. It is currently published partially, and the newest issue is volume 80, which was published in 2022 and records 35 species of Ruaceae in Bangladesh [45].

### 3.4. Bhutan

#### 3.4.1. *The Flora of Bhutan*, 3 Volumes, 9 Fascicles (1983–2002)

The *Flora of Bhutan* (three volumes, each with three parts, and, in total, nine fascicles) was completed during 1983 to 2002. It was mainly contributed by A.J.C. Grierson and D.G. Long, as well as by H.J. Noltie, N.R. Pearce, and P.J. Cribb [54,55,56,57,58,59,60,61,62]. The geographical range covered in this work not only included Bhutan, but also Sikkim and Darjeeling. There are, in total, 5985 taxa recorded with respect to the flora, including species, varieties, and subspecies. Following the description of each species, the information on distribution, ecology, and altitude is given. These significant recordings of flora do not simply provide the integrated classification of the flora of Bhutan, but also enumerate the history of botanical exploration of Bhutan and Sikkim in detail; further still, it also provides other useful information, such as the geographical outline and botanical bibliography of Bhutan.

#### 3.4.2. *Weeds of Bhutan* (1992) 

In this publication, there are illustrations of 187 species. This book was published, on behalf of the Royal Government of Bhutan, by Chris Parker in 1992. The publication is predominantly focused on the weeds of annual and perennial crops. However, certain parasitic weeds of forestry and some wild plants, which are likely poisonous to livestock, are also included. Local names, Latin names, characteristics, distribution, and importance are all listed in the book [133].

### 3.5. India 

#### 3.5.1. *The Flora of British India*, Vol. 1–7 (1872–1897)

*The Flora of British India* is still the most important record of the flora of India due to the revised *Flora of India* being far from completion [11]. Approximately 10,200 species (170 families, 2070 genera) out of the 14,300 species (171 families, 2325 genera) recorded in this publication concern flora that are located in the present boundaries of India.

#### 3.5.2. *Flora of India* (1997-)

The project for a new version, i.e., the *Flora of India*, has also been conducted by the BSI, and several volumes have already been published (the first three in 1997) and consist of more than 6000 pages of printed treatments, covering around 4500 taxa belonging to 92 families. Additionally, the *eFlora of India* is now available online (http://efloraindia.nic.in/efloraindia/homePage.action (accessed on 13 February 2023)). However, when this work will be completed is unknown. 

#### 3.5.3. *Flowering Plants of India: An Annotated Checklist* (2020) 

This book was published by the Botanical Survey of India in October 2020 [134]. It provides exhaustive information on 21,558 taxa of angiosperms that are located in India, under 268 families and 2744 genera. 

### 3.6. Maldives 

#### 3.6.1. *Trees and Shrubs of the Maldives* (2007) 

This book, by Selvam, contains records of only 100 woody plant species in Maldives [114]. However, the information on each species is quite detailed, including scientific names, synonyms, common names, local names, descriptions, uses, ecology, etc. There are also several photos for each species to show the habit, leaves, flowers, or fruits; background information about Maldives is also given in the book.

#### 3.6.2. *Common Plants of Maldives* (2016)

Compared to the publication mentioned above, this book—which was contributed by Sujanapal and Sankaran—contains the details of 270 vascular plants species in Maldives [115]. The given information for every species is similar, but the information about threat and damage is included. Additionally, the photo quality is better than the former.

#### 3.6.3. *Maldives Materia Medica Database* (http://www.biodiversity.com.mv/eng/, 10 October 2016)

There is also a database on biodiversity information, the *Maldives Materia Medica Database* (http://www.biodiversity.com.mv/eng/ (accessed on 13 February 2023)), which is available online. It includes more than one thousand records of plants in Maldives; however, many of these plants were introduced to the region and cultivated.

### 3.7. Nepal

#### 3.7.1. *Annotated Checklist of the Flowering Plants of Nepal* (2000)

The book version of the *Annotated Checklist of the Flowering Plants of Nepal* was published by the Natural History Museum of London in 2000 [122]. There is an updated version of this book that is available online. The list contains nearly 6500 flowering plants and ferns of Nepal. Currently, the available integrated inventory of wild plant diversity of Nepal is found in the ‘*Annotated Checklist of the Flowering Plants of Nepal*’ by Press and al., which is available in both solid book and online digital media (http://www.efloras.org/flora_page.aspx?flora_id=110, last access date 13 March 2023) formats. However, Nepal still does not have a completed national flora record to date. Having said this, the Flora of Nepal Project is currently being conducted, under an international consortium of partners headed by the Royal Botanic Garden Edinburgh (RBGE). Subsequent versions of the guidelines and supporting documents will be available via the Flora of Nepal website (www.floraofNepal.org, accessed on 13 February 2023).

#### 3.7.2. *Flora of Nepal*, Vol. 1–10. (2011-)

Volume 3 is the first of 10 volumes of the *Flora of Nepal*. It was published in 2011 [135] and documents 600 species in 21 families, from Magnoliaceae to Rosaceae. However, the time of completion of this flora work is still unknown.


*An Enumeration of The Flowering Plants of Nepal, Vol. 1–3. (1978–1982)*


This book is the source for preparing the list of endemic flowering plants of Nepal [118,119,120]. 

### 3.8. Pakistan

#### 3.8.1. *Flora of Pakistan*, Vol. 1–224 (1970–2020)

This book was published in a single-family volume, and at the beginning of each volume, there is a map showing the area covered by the journal, including the Gilgit and Ladakh regions. Originally known as the *Flora of West Pakistan* [124], it was renamed the *Flora of Pakistan* from volume 132 in 1979 onward [136]. These books are, in fact, the core of the *Flora of Pakistan* and provide an overview of not only the physical geography, but also the history of collection and research. The book has 224 volumes containing details of over 6000 indigenous flowering species [136].

#### 3.8.2. The Online Flora of Pakistan (http://www.efloras.org/flora_page.aspx?flora_id=5, Last Access Date 13 March 2023)

This eflora online database is based on and includes all the information that was published in the physically published *Flora of Pakistan*, which is the first modern flora record of more than 6000 species in South Asia.

### 3.9. Sri Lanka 

#### 3.9.1. *A Check List of the Flowering Plants of Sri Lanka* (2001) 

This list contains all the species that were recorded in the two new and old versions of the *Handbook to the Flora of Ceylon* [128,129], including some other works. This checklist includes 4143 species, 1522 genera, and 214 families. More than 20% of the species are local to Sri Lanka. The major limitation of this checklist, however, is that it is without distribution information for the species.

#### 3.9.2. *The Illustrated Field Guide to the Flowers of Sri Lanka* Vol. 1 (2008)

More than 1000 of different plant species that were found in Sri Lanka are depicted in this publication, and more than 200 related species are described in short [137].

#### 3.9.3. *The Illustrated Field Guide to the Flowers of Sri Lanka* Vol. 2 (2015)

Volume 2 of this publication contains 1060 species, of which 963 were not included in the first book, and were photographed throughout the whole of Sri Lanka, including the northern and northeastern regions of the country [138]. 

#### 3.9.4. *The Illustrated Field Guide to the Flowers of Sri Lanka* Vol. 3 (2019)

This third volume deals with 1067 plant species, of which 967 were not described in volume 1 and 2. Many of them are rare, and some are most likely new to the flora of Sri Lanka or possibly even to science [139]. 

## 4. Taxonomy Sources Available Online

In the last decade, a myriad of digitization initiatives, combined with the growth of computer and information technologies, have yielded a growing stream of data flowing from natural history collections (Table 1) and publications (Table 2) into aggregators [140]. Continually updated databases are replacing traditional methods of written flora records [141]; This can improve the way floristic data are assembled, circulated, reviewed, and accessed [142]. Such online databases have been established in South Asia and are outlined below.

These taxonomy sources provide valuable information for researchers, conservationists, and others who are interested in botanical diversity. Here, we extracted the information of native plants distributed in South Asia from the Catalogue of Life (COL, www.catalogueoflife.org, last access date 13 March 2023), which is an authoritative and comprehensive database of species names and information, including details on the distribution, ecology, and taxonomy of flora (COL, 2020). In addition, there is also the Global Biodiversity Information Facility (GBIF) (https://www.gbif.org/, last access date 30 December 2020), which provides access to over 2.2 billion biodiversity records (GBIF, 2020). The number of vascular plants for each country is different in the different databases. For example, the number of species in India in the GBIF is over 21,000, while in the COL, less than 14,000 species are recorded. We counted both names (at the species level) and specimens (with names at the species level) from South Asia in the COL and GBIF. The results are detailed in Table 3.

The *Flora of India* (www.efloras.org/flora_page.aspx?flora_id=110, last access date 13 March 2023) provides detailed information on over 15,000 species of plants, including descriptions, over 60,000 images, and distribution maps.

The *Database of Plants of Indian Subcontinent* (https://efloraofindia.com/, last access date 13 March 2023) has the largest database on the Internet on Indian flora, with more than 14,000 species along with more than 400,000 images.

The *Indian Medicinal Plants Database* (http://www.medicinalplants.in/, last access date 13 March 2023) incorporates 7263 botanical names of Indian medicinal plants. These botanical names have been correlated with more than 150,000 vernacular names in 10 different languages of India. It also includes >5000 authentic images of Indian medicinal plants, which are duly linked to specific botanical entities.

The *Flora of Sri Lanka* (https://www.floraofsrilanka.com/, last access date 13 March 2023) contains a total of 2278 plant species belonging to 1764 genera and 253 families. It provides in-depth information on the taxonomy, distribution, and ecology of these plants, along with high-quality images and descriptions.

The *Ayurvedic Medicinal Plants of Sri Lanka* (http://www.ayurvedicmedicinalplantssrilanka.org/, last access date 13 March 2023) lists 1373 species with scientific information on various species of plants, including their scientific names, common names, and uses in traditional Ayurvedic medicine.

## 5. Discussion

South Asia, which is better identified as the Indian subcontinent, is a natural biogeographic region that is isolated from the remaining Asian landmass by a nearly continuous mountain chain on its northwest, north, northeast, and eastern sides. The climate ranges from the tropical through to the subtropical, and the montane through to the temperate areas are in different parts, but these areas are governed by monsoons, which cause large spatial and temporal variations in precipitation. The vast territory of the region has a complex geographical and dramatic climate, which allows it to house a large diversity of plants, including certain species with high economic and medicinal value. Botanical exploration in the region has been carried out since the 17th century, resulting in the collection of a great deal of specimens, floras, checklists, and other kinds of publications. Most of these constitute a solid foundation for future botanical research in South Asian countries, which also includes the MAP project.

At present, the progress of floristic research in these eight countries is uneven. The flora of several countries is just beginning and has not yet been completed, such as the case for the *Flora of India* and the *Flora of Nepal* [135]. Moreover, the *Flora of Afghanistan* has been published for 67 years [41]. In the years since its publication, many new taxa and occurrences have been described and reported [143,144]. In addition, with the development of molecular biotechnology, the family and genus positions of certain species have also changed, and the plant classification systems in these eight countries also need to be updated with the recent publication of the APG system [145].

Today, biodiversity information databases have promoted rapid development at the global, regional, and national levels, such as *the Global Biodiversity Information Facility* (GIBF, https://www.gbif.org, last access date 13 March 2023), *the Integrated Digitized Biocollections* (iDigBio, https://www.idigbio.org/, last access date 13 March 2023), and *An integrated Big BioData Infrastructure for CASEarth* (BioONE, https://www.bio-one.org.cn/, last access date 13 March 2023). According to our incomplete statistics, there are 151 books that are devoted to the flora of different regions of South Asia. It is very difficult to clean the data while integrating taxonomic changes and heterogeneous data from multiple sources. However, by taking Pakistan as an example, the *Flora of Pakistan* records ca. 4494 taxa, the database of the COL records ca. 4874 taxa, and the specimens of the GBIF records belong to ca. 4456 taxa; however, the number of the same taxa that are shared by these three sources is only 1736. This indicates that all current data sources have certain deficiencies. Further analysis also shows that there are certain reasons for these deficiencies. Currently, the data are still greatly uncompleted. The records of floras or lists are often lacking with respect to ferns, while the GBIF data cover a very limited proportion of the taxa of the whole country. Moreover, the number of the proportion does not exceed 40% in our preliminary analysis. The different classification systems in different data sources result in a great deal of synonymous names for the same taxa, which thus results in an illusory high number of taxa for certain countries. The records of introduced plants (either cultivated or naturalized) not only create an illusory high number of taxa for certain countries, but may also cause erroneous analysis in phytogeography. However, currently, these types of records are still presented in all data sources, especially in the GBIF data. Therefore, to obtain comprehensive, accurate, and reliable data of plant catalogs in South Asian countries, we need to collect more sufficient data, unify the classification systems, eliminate synonyms, and distinguish between native plants and introduced plants in further works.

## 6. Conclusions and Future Directions

Although plant research work in South Asia started relatively early and has a good foundation in specimen collection and book publishing, it is also full of challenges. It requires several approaches to support scientific collaborations. Relatively speaking, the economy is relatively backward; furthermore, cooperation between countries is relatively limited in the region. Therefore, it is necessary to establish a regional mechanism that is similar to that of the MAP project to call on more institutions and organizations to join us in order to understand the ‘what’ and ‘where’ issues at the regional scale in South Asia. In order to better guide biodiversity conservation in the region, the following work is very important: (1) establishing a real-time updated species list at the regional scale; (2) integrating and cleaning data from different sources to establish a database of plant distribution; (3) making full use of institutions—such as the South Asian University, the working group at the Global Young Academy called Science Diplomacy, and The Cereal Systems Initiative—in order to cultivate more local scholars, as well as to promote plant research and the development of biodiversity conservation in South Asia.

## Figures and Tables

**Figure 1 plants-12-01617-f001:**
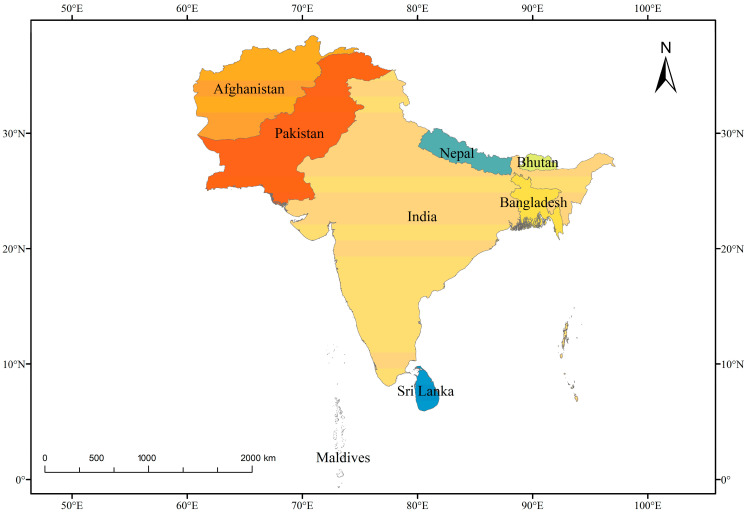
The countries of South Asia.

**Table 1 plants-12-01617-t001:** Summary of the herbaria in each country of South Asia.

Country 2021	Herbaria 2021	Specimens 2021	Staff and Associates
Afghanistan	2	33,609	3
Bangladesh	3	139,000	15
Bhutan	1	15,000	6
India	116	5,784,612	441
Nepal	2	1,850,000	24
Pakistan	29	681,670	84
Sri Lanka	8	335,689	7
Maldives	0	0	0

**Table 2 plants-12-01617-t002:** The approximated numbers of taxa in South Asian countries.

No.	Country	No. of Taxa	Floras
1	Afghanistan	5261	*Vascular Plants of Afghanistan: an Augmented Checklist* (2013) [132]
2	Bangladesh	3470	*Encyclopedia of Flora and Fauna of Bangladesh*, Vol. 5–12 (2007–2009) [46,47,48,49,50,51,52,53]
3	Bhutan	5985	*Flora of Bhutan*, 3 volumes, 9 fascicles (1983–2002) [54,55,56,57,58,59,60,61,62]
4	India	21,558	*Flowering Plants of India: An Annotated Checklist*, Vol. 1–3 (2020) [134]
5	Maldives	270	*Common Plants of Maldives* (2016) [115]
6	Nepal	6500	*Annotated Checklist of the Flowering Plants of Nepal* (2000) [122]
7	Pakistan	6000+	*Flora of West Pakistan,* vols. 1–131(1970–1979); *Flora of Pakistan*, vols. 132–224 (1980–2020) [124,136]
8	Sri Lanka	4143	*A Check List of the Flowering Plants of Sri Lanka* (2001) [129]

**Table 3 plants-12-01617-t003:** Statistic numbers for the vascular plants of South Asia from the GBIF (2020).

Countries	No. of Species (COL)	No. of Species (GBIF)	No. of Occurrences (GBIF)
**Afghanistan**	4257	6576	505,750
**Bangladesh**	2932	2166	22,275
**Bhutan**	3271	2931	11,398
**India**	13,897	21,229	347,023
**Maldives**	286	259	861
**Nepal**	6301	7169	74,526
**Pakistan**	4874	4456	82,603
**Sri Lanka**	4051	6078	61,222

## Data Availability

The raw data that support the findings of this study are from public databases, such as the GBIF (https://www.gbif.org/ accessed on 13 February 2023).

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
