# Peer review of "Mapping Asia Plants: Historical Outline and Review of Sources on Floristic Diversity in South Asia"

_plants, 2023, doi:10.3390/plants12081617_

Round 1
Reviewer 1 Report
This is an interesting and useful paper, but you may improve this article to publish in this journal. Otherwise, I have a lot of recommendations to increase the quality of your paper. Be careful with the writing and mistakes.
There is a keyword repeated in the article title. The keyword is “South Asia”. In order to increase the visibility of your paper I recommend changing this keyword. If you change it by other keyword, you will increase the probability that your paper could be found by future readers when they look for your paper in some databases like Scopus for example. If you repeat the same words in the article title and in keywords, less people could find your work. So, you must think about the visibility of your research.
Line 19. Just before “Afghanistan” you must write a space.
Line 32. You must write a space just before a reference. This is a very common mistake in the whole manuscript. Please, fix this tiny mistake and the quality of this paper will increase considerably.
Line 58. Just before the following sentence “The region is also…” you must write a space.
Line 61. Just before the first comma in this line you must delete the space.
Line 61. Just before “deciduous” you must delete the space.
Line 79. Just before “Then, the classic history…” you must write a space.
Line 140. Just before “H. Rechinger” you must delete the space.
Line 154. Just before “Their publications can be found…” you must delete the space.
Lines 156-160. The whole paragraph is quite confusing. Please, rephrase this paragraph because there are a lot of concepts and lines doubled.
Line 181. You must write a space just before the years 1933 and 1950.
Line 228. You must delete the space just at the end of the link.
Line 235. You must write a space just before “Wallich”.
Line 236. You must write a space just before the year 1820.
Line 258. You must write a space just before “Hara”.
Line 260. You must write a space just before “Suwal”.
Line 260. You must write a space just before the year 1960.
Line 280. You must write a point just after the letter “B”.
Line 311. You must delete the space just before the letter “D.”.
Line 333. You must write a space just before the number 1086.
Line 348. The word “Bandgladesh” is wrong written. please, delete the letter “d” between the letters “n” and “g”.
Line 371. Please, delete the space just before the number 300.
Line 379. Please, delete the space just before the bracket.
Line 473. The number 4,00,000 is quite confusing. Which number is exactly? 400,000 or 4,000,000. It has no sense.
Line 477. The number 1,50,000 is quite confusing. Which number is exactly? 1,500,000 or 150,000. It has no sense.
Line 509. Please, delete the space just after the gbif link.
This paper has no Matherial and Methods and no Conclusions. Please, fix it.
As well this article needs a map. Please, fix all the tiping mistakes. I have found a lot of them but maybe there are even more.
Otherwise, the authors adequately developed the Introduction, presenting the problems but you must write explicitly the objectives of this paper.
The authors are to be congratulated for the results obtained in this article.
Reviewer 2 Report
Review Report on
Historical outline and review of sources on floristic diversity in
South Asia
I read the manuscript “Historical outline and review of sources on floristic diversity in South Asia” submitted to the journal “plants.” The manuscript seems to be originated because of Mapping Asia Plant (MAP) project. On the surface, the manuscript title sounds okay. However, the manuscript is nothing more than a poorly written report. Therefore, I recommend that editors reject the manuscript. The reason behind this recommendation is that the scientific contribution of the manuscript is insufficient; however, after careful conceptualization in terms of content, presentation, and articulation may be published as a report.
To justify why the manuscript deserves rejection, I have provided some comments until section 2: Some key points that I want to highlight are:
1) Manuscript does not show a distinction between Southeast Asia (SEA) and South Asia; terms are used interchangeably even after selected countries (8) in SEA are defined in the introduction. This creates confusion for international audiences. This is the blunder from the authors. The first word in the abstract is an example of this type of mistake.
2) Poor grammatical structures for academic works: negligence of single space, improper commas, and poor choice of sentence structures and grammar throughout the manuscript.
3) No clear objectives as to why the research was carried out. Depending on the country’s economy and research priorities, specimens and online database creation can vary by country. As long as work has been done, I don’t think the documentation in English would make any difference, as the nomenclature system is standardized worldwide. However, if countries standardized the work in English, that would be Ideal. Also, check the last paragraph of the discussion; it seems out of context. Similar issues have never been raised before in the manuscript.
4) Factual errors: I know that Bangladesh and Pakistan were not independent countries before 1947; they were part of India. Authors have listed that publications for Pakistan before 1947 exist, which creates confusion, especially since the authors failed to mention that Pakistan as a country existed only after 1947. Further, the authors maintained that Kashmir is part of Pakistan, and this is a political issue that should be avoided as far as possible.
In the overall introduction of South Asia (lines 76-84), only the issue of India is mentioned. This highlights that the manuscript was not well thought out. Moreover, the issues of the publication are repetitive.
Other few specific comments can be found in the pdf file (plants_2251938-peer-review-R1.pdf).

Reviewer 3 Report
Dear authors,
I find your effort to review the history of floristic explorations of South Asia and report a bibliographic compendium of published floras from this region potentially very useful for botanists worldwide. Amount of your work invested in this study is impressive. Unfortunately, it cannot be published in its current form of presentation. You did not manage reporting this study in a formally acceptable way. Lists of references in the manuscript are a complete mess. You can keep some order in reporting references to different floras and floristic publications for some of the countries, if the numbers of references are low for a country, like those for Bangladesh or Maldives. Keeping an order for numerous publications, like those from India, seems to exceed your capacities in this study. It is hard to follow strictly your basic intention of separating references to historical and floristic literature. The result is a mess of intermingled different styles of references. Some sources are reported by using the family name of the first author and year of publication, while others are reported by using titles of floras and numbers of volumes or years of publication. Besides, some sources are mentioned in the text but not reported in the list of references. For example, the references by year of publication to Rebecca Pradhan and her account of Bhutanese rhododendrons, or to APGIV paper. Moreover, citation of references in the formal (numbered) list at the end of the manuscript is hardly formal. What do the first author names in references 9,10, 12, 14,17,26, 28, 30, 33 mean? Are these single letters supposed to represent abbreviated family names of authors? I am sorry, but this way of reporting literature sources cannot be accepted. I recommend a considerable re-formatting of this study, so that each literature source should be properly cited and could be easily found in the reference list. The Reference list itself should follow the standards of the journal, to which the manuscript is submitted.
I would like to provide some minor comments on Abstract, so that you could improve its presentation for further submissions.
L. 11-12: Any listings should be ordered according to a rule. Names of countries should be ordered alphabetically or geographically. Alphabetical ordering should be preferable.
L. 18: What do you mean by the “wider political coverage”? Probably, your meaning is wider geographical coverage by the British botanists, is not it? Please clarify.
L. 20. Please replace point by comma, otherwise the second sentence is logically incomplete.
L. 21-22. It is unlikely that flora of Bhutan (5985 taxa) is so much larger than flora of India (4500 taxa). Besides, the round numbers (3100, 4500, 600, 6000) are unlikely to be correct also. Obviously, you mean approximate estimates. Please clarify both issues.
There are numerous additional flaws across the text. I am sorry, but currently I refuse reporting these flaws to you.
Best regards,
IB
Reviewer 4 Report
This article is very interesting because it contains information that few readers know. South Asia is extremely rich in flora, and the study of this wealth is primarily founded on the achievements of the first explorers. That is why I consider it worthy of publication. However, I think that some improvements would make the content better structured and more convenient for the readers. I would suggest rearrangements which will not change the content, but will make easily get data country by country.
I enjoyed the reading but I was curious about the sources of information about the history of floristic research in South Asia. There is a lot of information that is clearly taken from various sources, but for which there are no citation. I mention only two examples here, but there are much more: “Frank Ludlow and George Sherriff made seven visits to Bhutan between1933 and1950 and collected over 6000 specimens (lines 180-182), or “Between 1960 and 1972, the University of Tokyo organized some botanical expeditions from eastern Nepal to Sikkim and Bhutan under the leadership of H.Hara who have made rich collections which are preserved in Tokyo.”(lines 257-259). Same is for Bhutan and Nepal where a lot of historical information is presented without mentioning any source. For authors convenience, they may introduce their historical data mining in a separate section. Please, include appropriate references in the manuscript.
The authors have preferred to separate the historical notes on floristic exploring form regional level floras and checklists. By my opinion, gathering the information for each country in one section will benefit for the readers. Country records could include both information together with the reference numbers of the local floras and checklists.
Floras and check lists and other literature sources mentioned in the text should be left as titles only and/or sited and included in the references as required in Instructions for authors. Here an example is given to get the idea for this suggestion: “Checklist of the Flowering Plants of Afghanistan was published by Podlech (2012), and is one of the basic reference sources for 'Vascular Plants of Afghanistan:an Augmented Checklist' published in 2013.”should be “Checklist of the Flowering Plants of Afghanistan is one of the basic reference sources for 'Vascular Plants of Afghanistan:an Augmented Checklist' [XX]”, or “Although there are accounts of travel journals and papers published by botanists visiting Bhutan, but significant floristic works don’t exist until the Flora of Bhutan” should be “Although there are accounts of travel journals and papers published by botanists visiting Bhutan, but significant floristic works don’t exist until the Flora of Bhutan [XX].” This way, many repetitions of the complete information of the literature sources in the text will be omitted. Same way the authors have build the text between lines 106-124. The authors probably consider the List of floristic literature included in the Appendix as a part of the results and this is true, but including them in the Reference list will keep this achievement.
In addition to the statements in lines 294-295, I think it would be useful to add information about existing herbariums in the mentioned countries. For example, I found in the Index Herbariorum that there is a legitimate herbarium in Pakistan, but I found no information about Bhutan.
Some minor remarks
Line 47 Please, change “higher plant species” with “vascular plant species”
Line 62 “vegetation types’ is better than “vegetations”
Line 101-102 Why some books are in italic and others are not?
Line 151 “…which were the results…” should be “which contains the results…”
Line 191 Please, explain what are RGOB and RBGE.
Line 209 The sentence needs a redaction.
Line 215 There are few / limited botanical works….
Line 282 Dehra Dun Forest Research Institute or elsewhere?
Line 293 Please revise “… was sent to Pakistan for a year to study the question…” to “…was sent to Pakistan for a year to study the likelihood…”
Line 310-319 Why there is no information about Flora of India (1997- ) ?
Line 382 Is the mentioned database linked to the book?
Line 514 What is COL record?
Please avoid repetition of South Asia so many times, e.g. Lines 52-60; 530-544.
In the reference list there are abbreviated author names. Please tape them in full, e.g. [14], [17], [22], [26], [28], [30], [33]..
Line 636-642 Literature about bryophytes is included in the reference list for India but in the text there is no comment about the mosses.
Line 728 correct to Flora of Misoram
Line 734 correct to Flora of Rajasthan
Please, check again the included in the text Internet links. I did not open most of them. The date of last accession should be also given
Round 2
Author Response
There is no comments and suggestions from Reviewer 2 in the Round 2.
Reviewer 3 Report
Dear authors,
You have done a good job in responding to my earlier criticism. The manuscript has been considerably improved. Besides, I do believe that your contribution on the contemporary state of botanical collections and literature in South Asia is important and timely. Your efforts convinced me, so I changed my attitude to your study. It probably can be accepted for publication in Plants. However, one substantial obstacle still remains. Language of the manuscript does not correspond to high standards of an international journal and must be improved. I suggest a professional editing service.
Best regards,
Igor Bartish, PhD
Author Response
A professional editing sevice has finished about this manuscript.

Reviewer 4 Report
I received the manuscript for the second round and consider it significantly improved. The authors have answered all comments. The most important improvement concerns the reference list and inclusion of required references in the text. However, some additional minor corrections will make the article better and my suggestions are listed below. I think that the manuscript could be published after considering these suggestions.
Line 225: Please omit ‘Rebecca Pradhan’
Line 244: Please check the English
Line 273-278: This sentence needs revision. It is too long and hard to get the message.
Line 282-283: the sentence seems incomplete
Line 290: British Museum (Natural History)?
Line 307-313: There is no reference for this statement
Line321: Lahore district flora (1936) – reference is missing
Line 338: Sri Lanka
Line 372 floristic feathers or features?
Line 398-403: reference is missing
Line 502-505: Please do not repeat in the text same data which is included in the Table 3.
in the Reference list: Please, permute 119 and 118; Please, start all country and region names with capital letters, e.g. lines 656,662,667,668,….675,682,…690,697,…725,727,788,802 and many others.
Round 3
Reviewer 2 Report
Dear Authors,
I failed to see a clean (without track changes) version of the manuscript and was not able to review it. I requested an editor to send me a clean version but never heard back. Please discard my ratings that are needed to submit this review.
Please follow along with the comments from other reviewers. Good luck!